# Decoration of CuO NWs Gas Sensor with ZnO NPs for Improving NO_2_ Sensing Characteristics

**DOI:** 10.3390/s21062103

**Published:** 2021-03-17

**Authors:** Tae-Hee Han, So-Young Bak, Sangwoo Kim, Se Hyeong Lee, Ye-Ji Han, Moonsuk Yi

**Affiliations:** 1Department of Smart Interdisciplinary Engineering, Pusan National University, Busan 46241, Korea; hanthee00@pusan.ac.kr (T.-H.H.); 201210107@pusan.ac.kr (S.K.); leopold00@pusan.ac.kr (Y.-J.H.); 2Department of Electronics Engineering, Pusan National University, Busan 46241, Korea; bso6459027@pusan.ac.kr (S.-Y.B.); shlee12@pusan.ac.kr (S.H.L.)

**Keywords:** gas sensors, CuO, nanowires, ZnO NPs, sol-gel, heterojunction

## Abstract

This paper introduces a method for improving the sensitivity to NO_2_ gas of a p-type metal oxide semiconductor gas sensor. The gas sensor was fabricated using CuO nanowires (NWs) grown through thermal oxidation and decorated with ZnO nanoparticles (NPs) using a sol-gel method. The CuO gas sensor with a ZnO heterojunction exhibited better sensitivity to NO_2_ gas than the pristine CuO gas sensor. The heterojunction in CuO/ZnO gas sensors caused a decrease in the width of the hole accumulation layer (HAL) and an increase in the initial resistance. The possibility to influence the width of the HAL helped improve the NO_2_ sensing characteristics of the gas sensor. The growth morphology, atomic composition, and crystal structure of the gas sensors were analyzed using field-emission scanning electron microscopy (FE-SEM), energy-dispersive X-ray spectroscopy, and X-ray diffraction, respectively.

## 1. Introduction

Metal oxide semiconductor (MOS) gas sensors have been studied owing to their semi-permanent performance, low price, easy manufacturing process, and high reactivity to gases [1,2,3]. MOS materials generally form adsorbed oxygen ions on nonstoichiometric surfaces at high temperatures (200–400 °C) and also a superficial conductive layer or a depletion layer depending on the type of MOS material. In MOS gas sensors, a chemical reaction between the adsorbed oxygen ions and the target gas causes a change in the thickness of the surface layer, thereby changing the resistance [4]. MOS gas sensors can comprise n-type and p-type materials that generate an electrical core-shell by adsorbing oxygen [5,6]. When oxygen ions are adsorbed on the surface of a p-type oxide semiconductor at 200–400 °C, a hole accumulation layer (HAL) and a high-resistance core are formed above and within the surface, respectively [7]. In general, when a p-type oxide semiconductor reacts to an oxidizing gas, the concentration of the surface holes increases and the resistance decreases [8]. Although the p-type MOS materials, such as CuO, Cr_2_O_3_, and NiO, show a high sensitivity to specific gases [7,9], they react less to oxidizing gases, compared to the n-type materials [10,11]. Therefore, there are less experimental results on p-type semiconductor materials than on n-type materials. However, the p-type materials are as important as n-type materials when used in gas sensors, such as complex gas sensor modules and electronic noses [7,12,13].

CuO is a p-type MOS material with a narrow band of 1.2−1.8 eV [14] and has excellent physical and electrical properties. CuO has been used in various applications, such as in solar cells [15], light emitting diodes (LEDs) [16], photocatalysts [17], and sensing materials [18,19]. However, CuO demonstrates a low sensitivity to oxidizing gases [7,20]. To improve its performance in gas reactions, various methods, such as doping and heterojunctions, have been studied [21,22].

In this work, we investigated the CuO/ZnO heterojunction. ZnO is an n-type material that improves the sensitivity of gas sensors in the form of core-shell [23] and decoration particles [24]. In addition, a high-efficiency gas sensor was realized through thermal oxidation in a tube furnace. This process is not as complex as atomic layer deposition (ALD) [25] and hydrothermal synthesis [26].

In this study, a heterojunction was realized by decorating CuO nanowires (NWs) with ZnO nanoparticles (NPs) to improve the NO_2_ gas detection characteristics. The CuO NWs were grown through thermal oxidation [27], and the ZnO NPs were used to form a CuO/ZnO heterojunction by employing a sol-gel method [28]. The sol-gel method is a simple yet highly efficient oxidization process that uses a sol-gel solution. The p-n heterojunction of the CuO and ZnO interfaces successfully improved the sensitivity of the gas sensors to NO_2_ gas, as well as their response and recovery time, selective properties.

## 2. Materials and Methods

### 2.1. Growth of the CuO NWs

The CuO NWs were grown through thermal oxidation of a Cu foil (99.98%, 0.25 mm thickness, Sigma-Aldrich) in a tube furnace. The Cu foil had a surface of 2 cm × 1 cm and, before undergoing thermal oxidation, it was cleaned in an aqueous solution of 1.5 M HCl for 1 min. Subsequently, it was cleaned with an ultrasonicator in acetone and deionized (DI) water for 10 min. The cleaned Cu foil was then placed on an alumina boat located in the middle of the tube furnace. The sample was heated to 600 °C (30 °C/min) for 3 h at atmospheric pressure. Once the thermal oxidation process was completed, the sample was cooled down in the tube furnace. 

A CuO NW suspension was prepared by ultrasonic dispersion of the thermally oxidized Cu foil in a DI water (2 mL) and isopropyl alcohol (1 mL) solution. The CuO gas sensor was realized by dropping the CuO NW suspension onto an Au electrode, and then dried on a hotplate at 120 °C. After drying the solvent in the CuO NW slurry, the gas sensor was annealed at 400 °C for 1 h in the tube furnace.

### 2.2. ZnO NPs Decoration on the CuO NWs Gas Sensor

In this study, the concentrations of the ZnO sol-gel solutions in the CuO NWs were 0.025, 0.05, and 0.075 M. Zinc acetate dihydrate (Zn(CH_3_COO)_2_∙2H_2_O, 99.999%, Sigma-Aldrich) was dissolved in 2-methoxyethanol. The amount of stabilizer (Ethanolamine) added was four times the concentration of the solute. The sol-gel solution was stirred at 75 °C and 500 rpm for 1 h on a hot plate, and the impurities were removed using a 0.2 μm syringe filter. The sol-gel solution was aged at 25 °C for 24 h and then coated with a spin coater at 1000 rpm for 30 s to form a CuO/ZnO heterojunction on the gas sensor. The CuO NWs coated with the ZnO sol-gel solution were dried for 10 min on a hot plate at 300 °C and then annealed for 1 h in the tube furnace at 500 °C. 

## 3. Results and Discussion

### 3.1. Material Analysis

#### 3.1.1. Field-Emission Scanning Electron Microscopy (FE-SEM) and Energy-Dispersive X-ray Spectroscopy (EDS) Analyses

Figure 1a shows the surface of a pristine CuO NW. The surface appears smooth, without any particles. The NWs had a length of 10–25 μm and a width of 300–600 nm. Figure 1b shows the surface of the 0.025 M ZnO/CuO NWs, in which small NPs were observed.

However, many NPs were observed on the surface of the CuO NWs decorated with 0.05 M ZnO NPs and 0.075 M ZnO NPs (Figure 1c,d respectively). These images show how many particles adhere to the surface of the NWs at different sol-gel concentration. Figure 1d shows that an excessive concentration of sol-gel solution can destroy the structure of the nanowires.

Figure 2 presents the results of the EDS analysis on the pristine CuO and the 0.05 M CuO/ZnO gas sensors. The Zn K peak was not detected in the pristine CuO sample, as depicted in Figure 2a. Figure 2b shows that the Zn K peak was detected at 3.78% in the 0.05 M CuO/ZnO gas sensor. As the concentration of the sol-gel solution increases, the weight of the detected Zn K peak increases (Table 1).

The EDS mapping images are demonstrated in Figure 3. It was observed that the NWs primarily comprised Cu and O. The Zn NPs were uniformly distributed over the entire area.

#### 3.1.2. X-ray Diffraction (XRD) Analysis

As shown in Figure 4, the crystal structures of the 0.05 M CuO/ZnO gas sensor were analyzed using XRD. The XRD patterns were indexed to CuO (ICSD number 87125) and to the hexagonal structure of ZnO (ICSD number 162843). This confirmed that the crystalline ZnO NPs synthesized through the sol-gel method were formed on the CuO NWs. The CuO NWs exhibited a high peak in the (11$\bar{1}$) and (111) directions at 35.558° and 38.748°, respectively. The ZnO nanoparticles exhibited a high peak in the (010), (002), and (011) directions at 31.777°, 34.426°, and 36.262°, respectively.

### 3.2. Effect of the CuO/ZnO Heterojunction on Gas Sensing

CuO and ZnO have band gaps of 1.8 eV and 3.2 eV, respectively. As shown in Figure 5a, when p-CuO and n-ZnO form a p-n heterojunction, the band curves towards the Fermi level, and a depletion layer is formed due to an electron and hole transfer process. Consequently, the HAL is partially suppressed, thereby increasing the resistance [29,30].

As shown in Figure 5b, the pristine CuO NW formed a HAL below the surface and a depletion layer inside. This is because of the oxygen ions adsorbed on the surface that was exposed to air at 200–400 °C. When the gas sensor was exposed to NO_2_, the oxidizing gas received electrons from the CuO NW and was adsorbed in the form of NO_2_^-^ ions on the surface. The corresponding loss of electrons in the CuO NW increased the concentration of holes and decreased the resistance. The radial HAL width modulation of the CuO NW heterojunction is shown in Figure 5c. The CuO/ZnO NW has a higher resistance than the pristine CuO NW owing to a partial carrier suppression along the radial direction of the ZnO NPs. Therefore, the oxidizing gas improved the gas response as it causes a significant change in the width of the conductive HAL [31].

### 3.3. Gas Sensing Performance

The gas sensing characteristics were evaluated in a cleanroom where the temperature and relative humidity were maintained at 20–25 °C and at 40%–50%, respectively. The gas measurement system consisted of 100 ppm cylinders of NO_2_, NH_3_, and CO; N_2_ (99.99%); a mass flow controller (MFC); and a pump (Figure 6). The gas sensor was located in the center of a 10 cm × 10 cm × 3 cm chamber. The MFC controlled the concentration of each gas by analyzing their flow rate. The pump discharged the gas to the outside and stabilized the pressure inside the chamber. The gas sensor, which was connected to electrodes that allowed current flow in the chamber, monitored the resistance. A change in the resistance indicated a reaction with the target gas. Before the reaction, the gas sensor was stabilized in an N_2_ atmosphere for 300 s and then exposed to NO_2_ gas for 300 s. After the reaction, the gas sensor was stabilized in an only N_2_ atmosphere for 300 s. The response of the CuO NW gas sensor to an oxidizing gas and a reducing gas was defined as R_a_/R_g_ and R_g_/R_a_, respectively, where R_a_ is the resistance of the gas sensor before the reaction and R_g_ is the resistance after the exposure to NO_2_ gas [32]. The sensitivity responses of CuO, 0.025 M CuO/ZnO, 0.05 M CuO/ZnO, and 0.075 M CuO/ZnO to 100 ppm NO_2_ gas were measured at 250 °C (Figure 7).

The experimental results listed in Table 2 show that the CuO/ZnO gas sensors have a better response, response time, and recovery time to 100 ppm NO_2_ gas at 250 °C, than the pristine CuO gas sensor. In particular, with the CuO/ZnO gas sensors, the response and recovery times were reduced from 60 s to 25 s and from 225 s to 150 s, respectively. The best response to NO_2_ was observed in the 0.05 M CuO/ZnO sensor. In fact, in this case the response was 4.1, which is 2.6 times higher than the response of the pristine CuO (1.58). The response to NO_2_ gas is worse in the 0.075 M CuO/ZnO. This is because, as shown in Figure 1d, an excessive concentration of ZnO covers most of the CuO, thus causing a competitive reaction (with CuO), affecting the sensitivity of the sensor.

The response and recovery times were defined as the time required to reach 90% of the saturation and the time-limited response of the sensor, respectively [33]. The ZnO NPs improved the recovery and response times of the CuO gas sensors because they caused an increase in the surface of the gas sensor, and therefore in the surface reacting to the target gas [34]. Consequently, the adsorption and desorption of gas molecules on the surface of the gas sensor are accelerated (spillover effect) [35].

Figure 8 illustrates that as the ZnO sol-gel concentration increases, the resistance of the gas sensor (R_a_) increases owing to a reduction in the width of the HAL. This demonstrates that different heterojunctions were formed depending on the concentration of ZnO.

To determine the optimal operating temperatures of the CuO and 0.05 M CuO/ZnO gas sensors, their performance was evaluated using 100 ppm NO_2_ at various temperatures. Figure 9a shows the resistance curves of the CuO and 0.05 M CuO/ZnO at 200–300 °C. The CuO decorated with ZnO NPs exhibited a significant change in the resistance; in fact, CuO/ZnO gas sensors have a higher R_a_ than the pristine CuO gas sensor. As the operating temperature increased, the resistance of both sensors decreased, which is a characteristic of a typical MOS device at high temperatures. As shown in Figure 9b, both the CuO and 0.05 M CuO/ZnO sensors exhibited the best response at 250 °C. In general, the CuO/ZnO heterojunction gas sensor exhibited a better response at all temperatures than the CuO gas sensor.

Table 3 and Figure 10 summarize the response to 100 ppm NO_2_, NH_3_, and CO gases at 250 °C. As mentioned before, the CuO NWs gas sensors are p-type semiconductor material. When they react to an oxidizing gas, such as NO_2_, their resistance decreases; thus, their sensitivity is expressed as R_a_/R_g_. However, when CuO NWs gas sensors react with reducing gases, such as NH_3_ and CO, their resistance increases; thus, their sensitivity is expressed as R_g_/R_a_. While the responses to NH_3_ and CO gases were similar, regardless of the type of heterojunction, a higher sensitivity to NO_2_ gas was observed in the same measurement conditions.

The response of the 0.05 M CuO/ZnO gas sensor to NO_2_ gas was measured at different NO_2_ concentrations (1−100 ppm) at 250 °C. Figure 11a shows that the response increased linearly (coefficient of determination, R^2^ = 0.99) as the NO_2_ gas concentration increased.

Table 4 briefly summarizes the response of the CuO-based gas sensors obtained in previous works. The response of the CuO NWs without the heterojunction is similar to what we obtained [36,37,38]. Moreover, our study shows better results on the response of CuO/ZnO heterojunction sensors to NO_2_ gas, compared to the results of [29] with the same NO_2_ concentrations.

## 4. Conclusions

We fabricated CuO NWs decorated with ZnO through thermal oxidation and a sol-gel method. The response, response time, and recovery time of the CuO-based sensors to NO_2_ gas exposure were significantly improved using a p-n heterojunction to control the width of the HAL. The sensitivity of the pristine CuO gas sensor improved by a factor of 2.6 using a 0.05 M ZnO heterojunction. In addition, the response and recovery times were reduced from 60 s to 25 s and from 225 s to 150 s, respectively. The CuO/ZnO heterojunction gas sensor possessed a higher sensitivity to NO_2_, which is an oxidizing gas, than NH_3_ and CO, which are reducing gases.

## Figures and Tables

**Figure 1 sensors-21-02103-f001:**
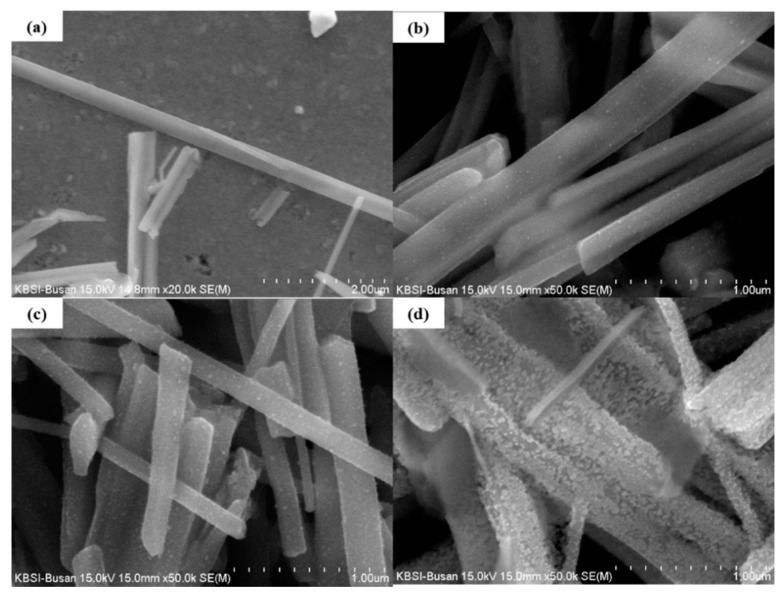
FE-SEM image of (**a**) pristine CuO, (**b**) CuO NWs decorated with 0.025 M ZnO NPs, (**c**) CuO NWs decorated with 0.05 M ZnO NPs, and (**d**) CuO NWs decorated with 0.075 M ZnO NPs.

**Figure 2 sensors-21-02103-f002:**
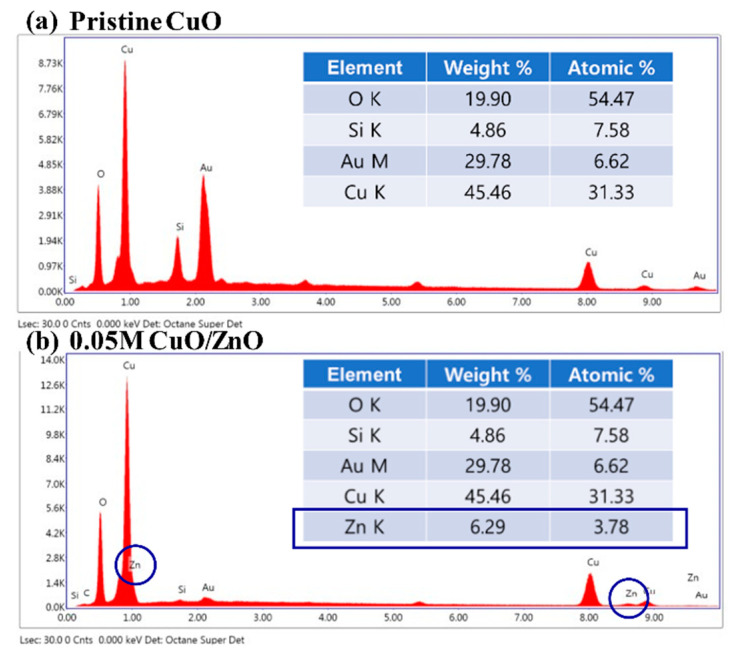
EDS analysis of (**a**) the pristine CuO, and (**b**) the 0.05 M CuO/ZnO.

**Figure 3 sensors-21-02103-f003:**
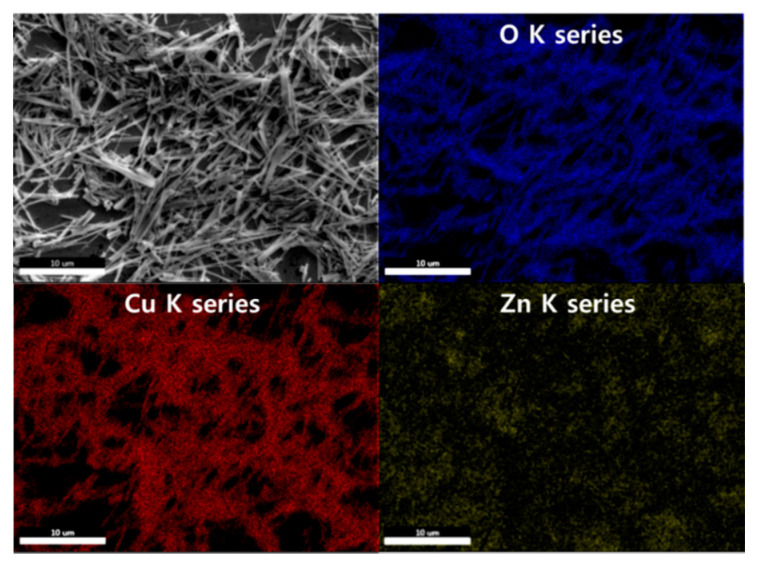
EDS mapping images of the 0.05 M CuO/ZnO gas sensor.

**Figure 4 sensors-21-02103-f004:**
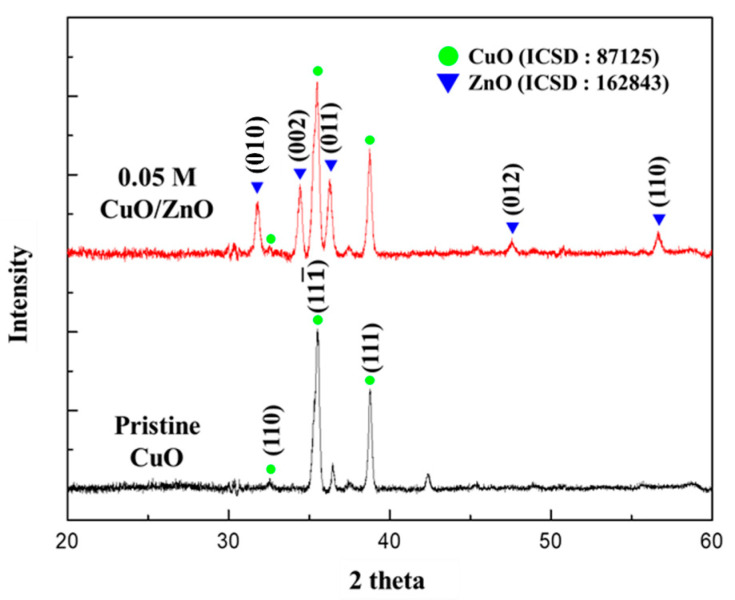
XRD analysis of the pristine CuO and the 0.05 M CuO/ZnO gas sensor.

**Figure 5 sensors-21-02103-f005:**
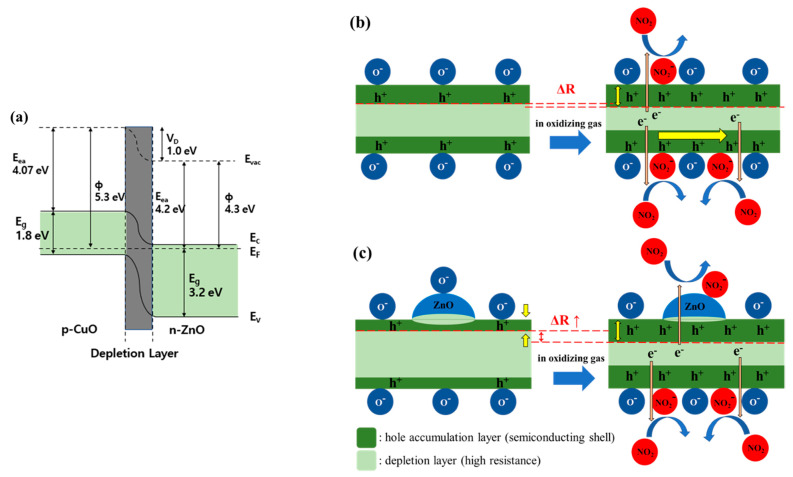
Schematic view of the effects caused by the CuO/ZnO heterojunction. (**a**) Diagram of the p-CuO and n-ZnO heterojunction band; (**b**) Reaction of a CuO NW to NO_2_ gas; (**c**) Reaction of a CuO NW with ZnO heterojunction to NO_2_ gas.

**Figure 6 sensors-21-02103-f006:**
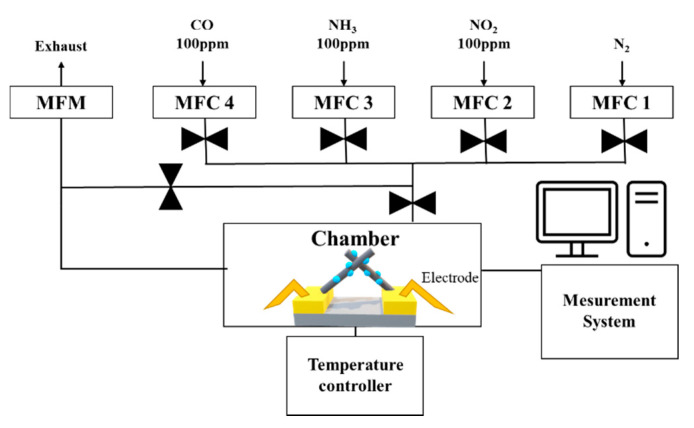
Architecture of the gas measurement system.

**Figure 7 sensors-21-02103-f007:**
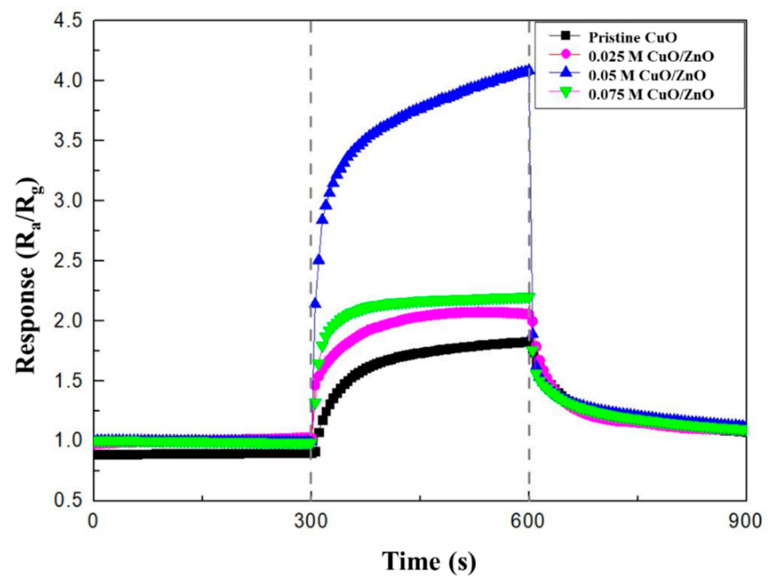
Response curves of the pristine CuO, 0.025 M CuO/ZnO, 0.05 M CuO/ZnO, and 0.075 M CuO/ZnO to 100 ppm NO_2_.

**Figure 8 sensors-21-02103-f008:**
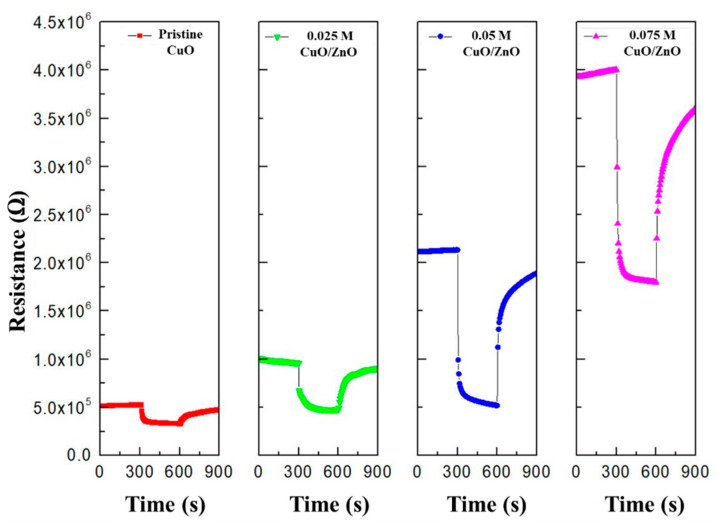
Resistance of the pristine CuO, 0.025 M CuO/ZnO, 0.05 M CuO/ZnO, and 0.075 M CuO/ZnO to 100 ppm NO_2_.

**Figure 9 sensors-21-02103-f009:**
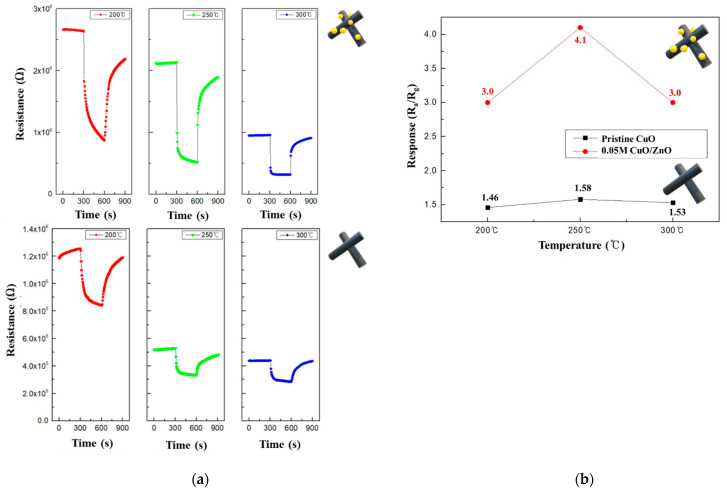
(**a**) Summary of the pristine CuO and 0.05 M CuO/ZnO responses to 100 ppm NO_2_ at 200-300 °C; (**b**) Response curves of the pristine CuO and 0.05 M CuO/ZnO to 100 ppm NO_2_ at 200–300 °C.

**Figure 10 sensors-21-02103-f010:**
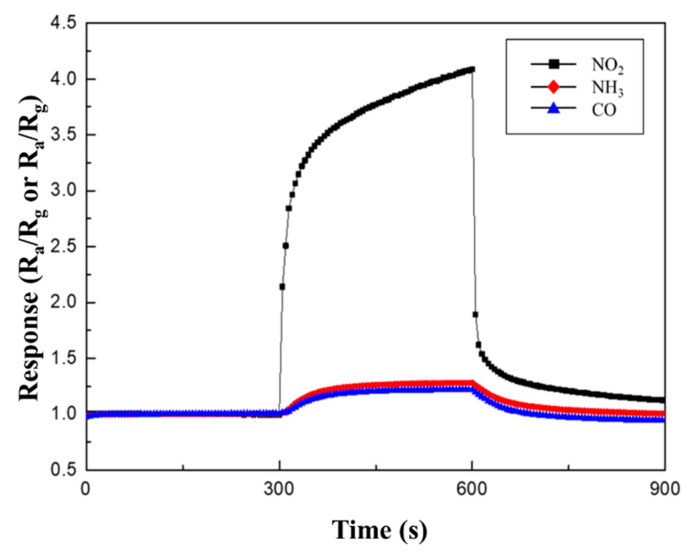
Response curves of 0.05 M CuO/ZnO to 100 ppm NO_2_, NH_3_, and CO gases at 250 °C.

**Figure 11 sensors-21-02103-f011:**
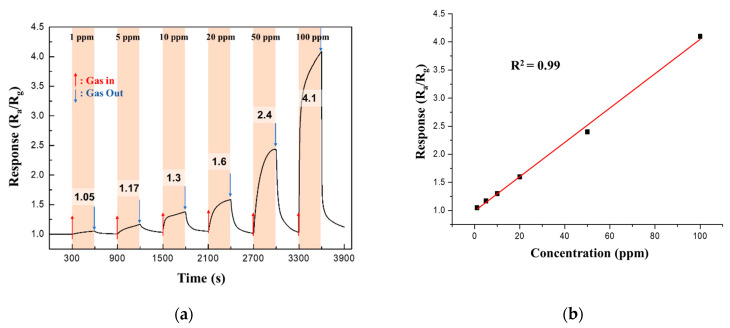
(**a**) Continuous response curves; (**b**) Response of the 0.05 M CuO/ZnO gas sensor to NO_2_ at different concentrations of NO_2_ (1−100 ppm) at 250 °C.

**Table 1 sensors-21-02103-t001:** EDS analysis of the Zn K peak for different concentrations.

	Pristine CuO	0.025 M CuO/ZnO	0.05 M CuO/ZnO	0.075 M CuO/ZnO
Zn K Weight % (wt.%)	-	1.17	6.29	8.66
Zn K Atomic % (at%)	-	0.75	3.78	5.66

**Table 2 sensors-21-02103-t002:** Response, response time, and recovery time of CuO-based gas sensors (with different sol-gel concentration) to 100 ppm NO_2_ gas at 250 °C.

Sample	Response (R_a_/R_g_)	Response Time(s)	Recovery Time(s)
Pristine CuO	1.58	60	225
0.025 M CuO/ZnO	2.0	65	180
0.05 M CuO/ZnO	4.1	25	150
0.075 M CuO/ZnO	2.22	20	180

**Table 3 sensors-21-02103-t003:** Response of the pristine CuO and 0.05 M CuO/ZnO to 100 ppm NO_2_, NH_3_, and CO gas at 250 °C.

	Pristine CuO	0.05 M CuO/ZnO
NO_2_	1.58	4.1
NH_3_	1.21	1.28
CO	1.19	1.18

**Table 4 sensors-21-02103-t004:** Brief summary of the response of CuO-based gas sensors.

Material	Structure	Method	Target Gas, Concentration (ppm)	Operating Temperature (°C)	Gas Response	Ref.
CuO	particles	Thermal decomposition	NO_2_, 100	200	1.5	[36]
CuO	nanowires	Thermal oxidation	NO_2_, 4	370	1.18	[37]
CuO	nanocubes	Polyol process	NO_2_, 3	300	0.4	[38]
CuO@ZnO	Core-shell	Two-step solution route	C_2_H_5_OH, 50	240	3.6	[39]
CuO-ZnO	Nanocomposites	Chemical method	NO_2_, 100	200	73 %	[29]
CuO/ZnO	nanowires	Thermal oxidation	NO_2_, 100	250	4.1	This work

## Data Availability

Not applicable.

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
