# Peer review of "Decoration of CuO NWs Gas Sensor with ZnO NPs for Improving NO2 Sensing Characteristics"

_sensors, 2021, doi:10.3390/s21062103_

Round 1

Reviewer 1 Report

This paper deals with the study of the effect of ZnO Nanoparticles (NP) deposited over CuO Nano wires to enhance the sensitivity and selectivity of an NO2 sensor. Although the proposal is very interesting, the authors do not establish the difference with those sensors reported in reference 3 by Miller et al. The proposal that the use of NP enhances the sensitivity and selectivity of the sensor is already established in that reference. Moreover, I have the following comments about the manuscript.

  1. The concentration detection was not dramatically  improved in the present paper comparing with the literature reports.
  2. In figure 1d, the NWs structure looks like collapsed, does this have something to do with the sensitivity decreasing for 0.075M case? Why the structure has this behavior? Please explain.
  3. The authors do not describe the gas measurement process. Was it in a static or dynamic system? How did you determined the gas concentration? Please describe.
  4. Depending on the answer of the last question, I do not think the response time (and recovery time) can be measured in a static system, if the measurement was dynamic, what were the measurement conditions? Gas flow, for instance. Please discuss these issues.
  5. The explanation of the sensitivity enhancement given in figure 5 is not clear, please improve the figure.
  6. The response curves presented in figures 6-10 do not show that the steady state had been reached, why the response behavior in function of concentration is linear in figure 10? I think it would be necessary to wait a longer time during these measurements, so that the steady state can be reached. Please explain.
  7. I think the response time decrement do not has a strong fundament. Please explain and comment.

Author Response

Dear Editors and Reviewers:
Thank you for your letter and for the reviewers’ comments concerning our manuscript entitled “ZnO NPs decorated CuO NWs gas sensor for improved NO2 sensing characteristics.” Those comments are all valuable and very helpful for revising and improving our manuscript, as well as the important guiding significance to our researches. We have revised some descriptions and added some experiments and discussions by studying comments carefully. The responds to the reviewers' comments are as follows:

#Reviewer

This paper deals with the study of the effect of ZnO Nanoparticles (NP) deposited over CuO Nano wires to enhance the sensitivity and selectivity of an NO2 sensor. Although the proposal is very interesting, the authors do not establish the difference with those sensors reported in reference 3 by Miller et al. The proposal that the use of NP enhances the sensitivity and selectivity of the sensor is already established in that reference. Moreover, I have the following comments about the manuscript.

1. The concentration detection was not dramatically  improved in the present paper comparing with the literature reports.

Response :  Thank you for your valuable question. In this experiment, a gas sensor with improved reaction and excellent productivity was fabricated through a relatively simple process, thermal oxidation and sol-gel method. Looking at the experimental details, it is possible to manufacture many gas sensors at once with only a CuO NWs slurry and a ZnO sol-gel solution. This advantage was added to the introduction *(p.2-line 51). It is not a dramatic improvement in terms of response, but I think it is meaningful that it is definitely improved compared to before ZnO heterojunction.

2. In figure 1d, the NWs structure looks like collapsed, does this have something to do with the sensitivity decreasing for 0.075M case? Why the structure has this behavior? Please explain.

Response : Thank you for your question. In the case of 0.075M sol-gel heterojunction, too many ZnO NPs were attached and the structure collapsed. Because of this, the sensitivity decreased at 0.075M. This indicates that the ZnO sol-gel concentration of this device is optimized at 0.05M. I added this *(p.3-line 101).

3. The authors do not describe the gas measurement process. Was it in a static or dynamic system? How did you determined the gas concentration? Please describe.

Response : Thank you for your question. The gas measurement process used a dynamic system. It was measured by controlling the flow rate of the target gas and N2 with the MFC (Mass Flow Controller) connected to the chamber. The contents of the gas measurement system have been added to the text *(p.6-line 150).

4. Depending on the answer of the last question, I do not think the response time (and recovery time) can be measured in a static system, if the measurement was dynamic, what were the measurement conditions? Gas flow, for instance. Please discuss these issues.

Response : Thank you for your comment. As with the answer above, the conditions and contents of gas measurement have been added to the text *(p.6-line 150). In addition, the definition of response time and recovery time has been added along with the reference *(p.7-line 176).

5. The explanation of the sensitivity enhancement given in figure 5 is not clear, please improve the figure.

Response : Thank you for your suggestion. The text and picture contents have been partially modified for readability *(p.5-Figure.5).

6. The response curves presented in figures 6-10 do not show that the steady state had been reached, why the response behavior in function of concentration is linear in figure 10? I think it would be necessary to wait a longer time during these measurements, so that the steady state can be reached. Please explain.

Response : Thank you for your question. It is difficult to measure by waiting until all reactions are saturated. Other references also frequently use time-limited measurements and this is the common measurement method (e.g. https://doi.org/10.1016/j.snb.2004.09.048, https://doi.org/10.3390/s40670095 etc). In the case of the reaction behavior of the concentration function, it is considered that they all react during the same measurement time and show a linear correlation.

7. I think the response time decrement do not has a strong fundament. Please explain and comment.

Response : Thank you for your question. In my opinion, the decrease in response time is due to the increase in surface area due to the heterojunction of the first ZnO NPs and secondly due to the spillover effect similar to that of the catalyst of ZnO NPs. These contents have been added to the text along with references *(p.7-181 line).

We have tried our best to improve the manuscript and made some changes in the manuscript. We appreciate for editor and reviewers’ work earnestly, and hope that the correction will meet with approval, however, if there are other questions, we are willing to revise it again.
Once again, thank you very much for your comments and suggestion.
Best regards,
Tae-Hee Han

Reviewer 2 Report

This paper describes the development of a chemiresistive type gas sensor for NO2 with improved response by forming n-p hetero junction. The material of copper oxide with zinc oxide is well characterised and sensing experiments are well performed. They are claiming an improved response but they need to compare their performance with the literature (in the form a table) before this work can be published. Furthermore there are no experiments with humidity or selectivity which is an ongoing issue with all chemiresistive sensors. Without these additional experiments and analysis of literature I do recommend publication.

Author Response

Dear Editors and Reviewers:
Thank you for your letter and for the reviewers’ comments concerning our manuscript entitled “ZnO NPs decorated CuO NWs gas sensor for improved NO2 sensing characteristics.” Those comments are all valuable and very helpful for revising and improving our manuscript, as well as the important guiding significance to our researches. We have revised some descriptions and added some experiments and discussions by studying comments carefully. The responds to the reviewers' comments are as follows:

#Reviewer

This paper describes the development of a chemiresistive type gas sensor for NO2 with improved response by forming n-p hetero junction. The material of copper oxide with zinc oxide is well characterised and sensing experiments are well performed. They are claiming an improved response but they need to compare their performance with the literature (in the form a table) before this work can be published. Furthermore there are no experiments with humidity or selectivity which is an ongoing issue with all chemiresistive sensors. Without these additional experiments and analysis of literature I do recommend publication.

Response : Thank you for your kind question and suggestion. First, the humidity and other gases selectivity test cannot be performed with the currently set up equipment, so the experiment will be conducted after setting up the equipment. And i really want this additional experiment and result. Second, I agree with your good suggestion and compare the results of other papers and summarize them in a table *(p.10-line 227).

We have tried our best to improve the manuscript and made some changes in the manuscript. We appreciate for editor and reviewers’ work earnestly, and hope that the correction will meet with approval, however, if there are other questions, we are willing to revise it again.
Once again, thank you very much for your comments and suggestion.
Best regards,
Tae-Hee Han

Reviewer 3 Report

The authors present a relative simple resistive gas detector that is comprised of ZnO NPs decorated CuO NWs. Measured resistance with respect to the baseline in the presence of NO2 gas is presented. Overall, I think the work needs to undergo major revisions before publication could be considered. My comments, questions and suggestions are listed below.

Even though concise, Introduction appears to be too limitative. In particular, a stronger background on the actual sensor fabricated with CuO NW decorated with ZnO will be encouraged. I recommend a few linking sentences to improve the flow and readability of the introduction section.

Which adsorption process of this gas sensor mainly belongs to? Physical adsorption or chemisorption? Since it has been mentioned in this manuscript, it will be good if it can be further discussed in the mechanism.

Section 3.1.1 is presented with poor details. It is just a description of SEM images, EDS and XRD analyses. Please evidence what is novel and what is not in the presented data. In some cases Figure 2 and Table I provide the same data…I suggest to use just a picture using the other two spectra.

The analysis of Figure 5 seems to be not well connected to the rest of the section. Please include evidence on where the data and the mechanism were retrieved.

Please include some details on the MOS structure with emphasis on the fabricated device, dimensions and electrical characteristics. Moreover, belonging to resistive gas sensors family, more details on the setup and the evaluation of the resistance is necessary.

How the data of Table 2 were evaluated? They seems to be indicative of an adsorption/desorption process.

What are the unit of data reported in Table 3? Is it Ra/Rg?

Not clear why on the y-axis of Figure 9 is reported Ra/Rg or Rg/Ra. Why is it equivalent.

The experiments only demonstrate the response of Specific gas . As a practical gas sensing application, this work should also present a more comprehensive discussion on its sensitivity and selectivity and reliability of the results. Moreover, what a discussion on a  real scenario application would be useful.

Even though authors claim the sensor improve the sensing characteristics, it failed to give comprehensive experiments to demonstrate the performance of gas sensors. In particular a comparison with other similar devices (e.g. 10.1016/j.apsusc.2018.07.112) or other kind of electrical gas sensors (10.3390/s20072143) should be provided.

Author Response

Dear Editors and Reviewers:
Thank you for your letter and for the reviewers’ comments concerning our manuscript entitled “ZnO NPs decorated CuO NWs gas sensor for improved NO2 sensing characteristics.” Those comments are all valuable and very helpful for revising and improving our manuscript, as well as the important guiding significance to our researches. We have revised some descriptions and added some experiments and discussions by studying comments carefully. The responds to the reviewers' comments are as follows:

#Reviewer

The authors present a relative simple resistive gas detector that is comprised of ZnO NPs decorated CuO NWs. Measured resistance with respect to the baseline in the presence of NO2 gas is presented. Overall, I think the work needs to undergo major revisions before publication could be considered. My comments, questions and suggestions are listed below.

1. Even though concise, Introduction appears to be too limitative. In particular, a stronger background on the actual sensor fabricated with CuO NW decorated with ZnO will be encouraged. I recommend a few linking sentences to improve the flow and readability of the introduction section.

Response : Thank you for your suggestion. We agreed to your opinion and wrote the introduction reinforced.

2. Which adsorption process of this gas sensor mainly belongs to? Physical adsorption or chemisorption? Since it has been mentioned in this manuscript, it will be good if it can be further discussed in the mechanism.

Response : Thank you for your question and suggestion. First, the metal oxide semiconductor gas sensor mainly chemically adsorbs the target gas. and contents on the adsorption process have been added to help understand the entire contents *(p.1-line 25, p.5-line 136).

3. Section 3.1.1 is presented with poor details. It is just a description of SEM images, EDS and XRD analyses. Please evidence what is novel and what is not in the presented data. In some cases Figure 2 and Table I provide the same data…I suggest to use just a picture using the other two spectra.

Response : Thank you for your suggestion. Some analyzes were added for each measurement. In cases of Figure 2 and Table 1, Since this experiment aims to compare the degree of decoration according to the molar concentration of ZnO, a comparison according to the molar concentration of decorated ZnO is necessary. Also, I want to keep the EDS graph and table together because there are only some overlapping contents. 

4. The analysis of Figure 5 seems to be not well connected to the rest of the section. Please include evidence on where the data and the mechanism were retrieved.

Response : Thank you for your suggestion. The location of Figure 5 was set in consideration of the reader from the front. If you have a suggestion of suitable location to move, I'd appreciate it if you could tell me. Also, the contents of the mechanism were explained with reference.

5. Please include some details on the MOS structure with emphasis on the fabricated device, dimensions and electrical characteristics. Moreover, belonging to resistive gas sensors family, more details on the setup and the evaluation of the resistance is necessary.

Response : Thank you for your suggestion. This gas sensor is not MOS structure. I'm sorry for my inaccurate expression. The MOS means Metal Oxide Semiconductor materials in this article. This gas sensor is a simple structure composed of Au electrode and MOS material *(section 2. Materials and Methods). Details on resistance evaluation have been added *(p.6-line 150, 158).

6. How the data of Table 2 were evaluated? They seems to be indicative of an adsorption/desorption process.

Response : Thank you for your question. As described above, a description of the resistance measurement evaluation was added *(p.6-line 150). In addition, definitions of reaction time and recovery time were added along with the reference *(p.7-line 176).

7. What are the unit of data reported in Table 3? Is it Ra/Rg?

Response : Thank you for your question. When the CuO gas sensor reacts with NO2, which is an oxidizing gas, the resistance decreases, and the response is expressed in Ra/Rg. On the contrary, when reacting with reducing gases NH3 and CO, the resistance increases and the response is expressed as Rg/Ra. *(p.6-line 158, p.9-line 211)

8. Not clear why on the y-axis of Figure 9 is reported Ra/Rg or Rg/Ra. Why is it equivalent.

Response : Thank you for your question. For a similar reason to the answer in #7, this is a method of expressing a response (e.g. 10.1016 / j.apsusc.2018.07.112).

9. The experiments only demonstrate the response of Specific gas . As a practical gas sensing application, this work should also present a more comprehensive discussion on its sensitivity and selectivity and reliability of the results. Moreover, what a discussion on a  real scenario application would be useful.

Response : Thank you for your suggestion.  The detection system in our laboratory consists of a setup of NO2, NH3, and CO. However, since these gases are typical oxidizing and reducing gases, they will react similarly to other gases. As additional equipment comes in, other gases will be tested. Reliability showed similar or slightly decreased responses up to 40 days. The actual scenario application will be useful in mass production or complex gas sensors as mentioned in the introduction.

10. Even though authors claim the sensor improve the sensing characteristics, it failed to give comprehensive experiments to demonstrate the performance of gas sensors. In particular a comparison with other similar devices (e.g. 10.1016/j.apsusc.2018.07.112) or other kind of electrical gas sensors (10.3390/s20072143) should be provided.

Response : Thank you for your suggestion. We agree with your opinion and summarize the comparison with other similar devices in a table 4 *(p.10-Table 4). As equipment is added, comprehensive experiments such as humidity, selectivity, and sensitivity will be conducted, and additional papers will be written.

We have tried our best to improve the manuscript and made some changes in the manuscript. We appreciate for editor and reviewers’ work earnestly, and hope that the correction will meet with approval, however, if there are other questions, we are willing to revise it again.
Once again, thank you very much for your comments and suggestion.
Best regards,
Tae-Hee Han

Round 2

Reviewer 1 Report

Although in the reviewed version of the manuscript, some improvement has been shown, I still have a concern about the gas response measurement system. Maybe a figure of the measurement setup can be added to clarify how the gas sensor responses were performed. Or if the system has been reported in a previous work, the authors can add the reference. If the measurement system is static or dynamic, it depends on the measurement chamber dimensions and the flow conditions, as well as the location of the sensor inside the measurement chamber. That is the reason why it is not clear to me the improvement in the response time. Please explain.

There are still many English problems, I think the manuscript must be reviewed and corrected by a native speaker.

Author Response

Dear Editors and Reviewers:
Thank you for your letter and for the reviewers’ comments concerning our manuscript entitled “ZnO NPs decorated CuO NWs gas sensor for improved NO2 sensing characteristics.” Those comments are all valuable and very helpful for revising and improving our manuscript, as well as the important guiding significance to our researches. We have revised some descriptions and added some experiments and discussions by studying comments carefully. The responds to the reviewers' comments are as follows:

#Reviewer

Although in the reviewed version of the manuscript, some improvement has been shown, I still have a concern about the gas response measurement system. Maybe a figure of the measurement setup can be added to clarify how the gas sensor responses were performed. Or if the system has been reported in a previous work, the authors can add the reference. If the measurement system is static or dynamic, it depends on the measurement chamber dimensions and the flow conditions, as well as the location of the sensor inside the measurement chamber. That is the reason why it is not clear to me the improvement in the response time. Please explain.

Thank you for your valuable suggestion. We agreed with your opinion and added a schematic of gas measurement system. And add dimension of chamber, location of sensor.

There are still many English problems, I think the manuscript must be reviewed and corrected by a native speaker.

I’m sorry for my english problems. I will upload the corrected manuscript as soon as possible.

We have tried our best to improve the manuscript and made some changes in the manuscript. We appreciate for editor and reviewers’ work earnestly, and hope that the correction will meet with approval, however, if there are other questions, we are willing to revise it again.
Once again, thank you very much for your comments and suggestion.
Best regards,
Tae-Hee Han

Reviewer 2 Report

The authors have made the required change for the manuscript by adding a table to compare existing copper oxide sensors, however it is not sufficient to just add the table and not have a discussion. They need to discuss and compare their responses with the ones listed in table. A few sentences on page 10 is recommended before publication.

Author Response

Dear Editors and Reviewers:
Thank you for your letter and for the reviewers’ comments concerning our manuscript entitled “ZnO NPs decorated CuO NWs gas sensor for improved NO2 sensing characteristics.” Those comments are all valuable and very helpful for revising and improving our manuscript, as well as the important guiding significance to our researches. We have revised some descriptions and added some experiments and discussions by studying comments carefully. The responds to the reviewers' comments are as follows:

#Reviewer

The authors have made the required change for the manuscript by adding a table to compare existing copper oxide sensors, however it is not sufficient to just add the table and not have a discussion. They need to discuss and compare their responses with the ones listed in table. A few sentences on page 10 is recommended before publication.

Thank you for your kind suggestion. We agreed with your opinion and added a few sentences (p.9-line 226).

We have tried our best to improve the manuscript and made some changes in the manuscript. We appreciate for editor and reviewers’ work earnestly, and hope that the correction will meet with approval, however, if there are other questions, we are willing to revise it again.
Once again, thank you very much for your comments and suggestion.
Best regards,
Tae-Hee Han

Reviewer 3 Report

I suggest to use same precision for data presentation. Moreover, error bar will be highly suggested as it give an estimation of repeatability of the evaluation.

In some figures the scale should be improved (Figure7 and 8). As the scale does not allow to see the signal with higher accuracy.

Even though not strictly required, more care on the figure would result in a more effective manuscript.

Author Response

Dear Editors and Reviewers:
Thank you for your letter and for the reviewers’ comments concerning our manuscript entitled “ZnO NPs decorated CuO NWs gas sensor for improved NO2 sensing characteristics.” Those comments are all valuable and very helpful for revising and improving our manuscript, as well as the important guiding significance to our researches. We have revised some descriptions and added some experiments and discussions by studying comments carefully. The responds to the reviewers' comments are as follows:

#Reviewer

I suggest to use same precision for data presentation. Moreover, error bar will be highly suggested as it give an estimation of repeatability of the evaluation.

Thank you for your valuable suggestion. But, there are not a few graphs that can be displayed with error bars. Also, since the variation of the error bar is not large, we were decided not to add.

In some figures the scale should be improved (Figure7 and 8). As the scale does not allow to see the signal with higher accuracy.

In Figures 7 and 8, I think it is more important to compare the gas sensor's stabilizing resistance Ra, rather than looking at the signal. That is why we unified the Y-axis in Figures 7 and 8.

Even though not strictly required, more care on the figure would result in a more effective manuscript.

Thanks for your kind review. A native speaker's proofreading and picture correction are in progress. I will upload it as soon as the modification is complete.

We have tried our best to improve the manuscript and made some changes in the manuscript. We appreciate for editor and reviewers’ work earnestly, and hope that the correction will meet with approval, however, if there are other questions, we are willing to revise it again.
Once again, thank you very much for your comments and suggestion.
Best regards,
Tae-Hee Han